# Experimental and Numerical Investigation of Flow Structure and Heat Transfer Behavior of Multiple Jet Impingement Using MgO-Water Nanofluids

**DOI:** 10.3390/ma16113942

**Published:** 2023-05-25

**Authors:** Tsz Loong Tang, Hamidon Salleh, Muhammad Imran Sadiq, Mohd Anas Mohd Sabri, Meor Iqram Meor Ahmad, Wan Aizon W. Ghopa

**Affiliations:** 1Centre for Energy and Industrial Environment Studies (CEIES), Faculty of Mechanical and Manufacturing Engineering, Universiti Tun Hussein Onn Malaysia, Parit Raja 86400, Batu Pahat, Johor, Malaysia; 2Department of Mechanical and Manufacturing Engineering, Universiti Kebangsaan Malaysia (UKM), Bangi 43600, Selangor, Malaysia

**Keywords:** nanofluids, heat transfer enhancement, CFD simulation, multiple jet impingement

## Abstract

Nanofluids have attracted significant attention from researchers due to their ability to significantly enhance heat transfer, especially in jet impingement flows, which can improve their cooling performance. However, there is a lack of research on the use of nanofluids in multiple jet impingements, both in terms of experimental and numerical studies. Therefore, further investigation is necessary to fully understand the potential benefits and limitations of using nanofluids in this type of cooling system. Thus, an experimental and numerical investigation was performed to study the flow structure and heat transfer behavior of multiple jet impingement using MgO-water nanofluids with a 3 × 3 inline jet array at a nozzle-to-plate distance of 3 mm. The jet spacing was set to 3, 4.5, and 6 mm; the Reynolds number varies from 1000 to 10,000; and the particle volume fraction ranges from 0% to 0.15%. A 3D numerical analysis using ANSYS Fluent with SST k-ω turbulent model was presented. The single-phase model is adopted to predict the thermal physical nanofluid. The flow field and temperature distribution were investigated. Experimental results show that a nanofluid can provide a heat transfer enhancement at a small jet-to-jet spacing using a high particle volume fraction under a low Reynolds number; otherwise, an adverse effect on heat transfer may occur. The numerical results show that the single-phase model can predict the heat transfer trend of multiple jet impingement using nanofluids correctly but with significant deviation from experimental results because it cannot capture the effect of nanoparticles.

## 1. Introduction

Liquid jet impingement is a promising cooling technique used in many applications, from manufacturing to electronic cooling, because it has one of the highest known single-phase heat transfer coefficients [1]. Furthermore, the multiple impingement jet has received more attention than the single impingement jet since using an array of high-speed liquid jets to strike the target surface has a more significant potential than a single large jet to provide a uniform surface temperature [2]. However, the effort of researchers to optimize the heat transfer performance of impingement jets is limited due to the poor thermal property of traditional coolants, such as water, ethylene glycol, and fluorocarbon liquid.

Consequently, nanofluid is considered to enhance the heat transfer capability after Roy et al. [3] discovered the benefit of nanofluid through their numerical analysis by applying this coolant in jet impingement cooling. The nanofluid is the base fluid dispersed with a size smaller than 100 nm that has higher thermal conductivity than the base fluid and is sufficiently small to avoid sedimentation and channel clogging.

Extensive studies of jet impingement using nanofluid have been performed in the past, mainly using water as base fluids since water is the most common coolant used in jet impingement, and they can be found elsewhere [4,5,6,7,8,9,10,11,12,13,14,15,16,17,18,19,20,21,22,23,24,25,26,27,28]. In addition, most of the experimental studies on water-based nanofluid jet impingement have focused on flat target surfaces placed vertically below the nozzle. The main parameters studied typically fall within ranges of 1 ≤ N_jet_ ≤ 81, 0.0001 ≤ ϕ ≤ 10, 2 ≤ s/d ≤ 8.75, 0.5 ≤ H/d ≤ 40, and 200 ≤ Re_d_ ≤ 88,000, which are summarized in Table 1.

From Table 1, each nanofluid impingement jet research’s heat transfer enhancement is reviewed in terms of the Nusselt number or heat transfer coefficient ratio of nanofluid relative to water, i.e., Nu_nf_/Nu_f_ or h_nf_/h_f_. Nanofluids enhance the heat transfer of impingement jet if the ratios are greater than unity. In contrast, the nanofluids deteriorate the performance of jet impingement if the ratios are lower than unity. Based on Table 1, despite the heat transfer enhancement, we can also notice that the nanofluids might cause heat transfer degradation for jet impingement. Therefore, conducting experimental research on jet impingement with nanofluids is crucial to develop a comprehensive understanding of the heat transfer mechanism involved in using this type of coolant. This understanding would contribute to maximizing the cooling performance and efficiency of nanofluids in jet impingement cooling.

The heat transfer enhancement of jet impingement using nanofluids can be explained by nanofluids’ thermal conductivity enhancement, nanoparticles’ thermal dispersion effect, and the intense bombardment of nanoparticles on the heating surface that induces a turbulent effect to reduce the thickness of the boundary layer. The thermal dispersion effect is the chaotic movement of the nanoparticles in the fluid that accelerates the energy exchange in the coolant, and this effect is intensified with smaller particles size and higher fluid temperature. These three factors have been used to explain the heat transfer enhancement of jet impingement using nanofluid by Li et al. [8], Zhou et al. [9], Lv et al. [15], Lv et al. [16], Barewar et al. [17], Kareem et al. [18], Sorour et al. [19], Amjadian et al. [20], Balla et al. [21], and Al-Zuhairy et al. [25].

However, the degradation of heat transfer for impingement jet using nanofluids is attributed to the formation of nanoparticles layer on the target surface, high viscosity of nanofluids, and the existing recirculating flow structure in jet impinging flow. Chang and Yang [13], Barewar et al. [17], and Balla et al. [21] pointed out that the boiling phenomenon in impingement jet causes the unstable nanoparticle residues to form a nano-sorption layer resulting in additional thermal resistance on the heating surface.

Furthermore, the high viscosity of nanofluids might overshadow the positive effect of enhanced thermal conductivity of nanofluids because the nanoparticle volume fraction is too high or too low, or excessive surfactants are used. For instance, Senkal and Torri [22] found jet impingement cooling with worse heat transfer performance when a high-volume fraction (6 vol%) is used. In contrast, Zhou et al. [9] discovered that the impinging jet heat transfer performance is poor if the low volume fraction is used due to the high viscosity brought by the surfactant, and Tie et al. [23] found that adding excessive surfactant can cause decrement in the heat transfer rate of jet impingement.

Another explanation for the degradation of heat transfer in nanofluid jet impingement is the presence of a recirculating flow structure. This concept was first proposed by Nguyen et al. [5], who suggested that there is a large recirculating fluid structure downstream of the radial flow in single jet impingement. The nanofluid is trapped in this region and heated up quickly due to its high thermal conductivity. As a result, the thermal boundary layer in this region reaches the surface, leading to a reduction in heat transfer. However, at that time, Nguyen et al. [5] did not confirm the exact flow field of the impinging jet flow since they only observed the flow using the naked eye.

Later, Senkal and Torri [24] investigated the multiple impingement jet arrangement in a polar array experimentally and numerically. They found that the heat transfer of nanofluids is degraded, and they proposed the same deduction as Nguyen et al.’s explanation since their numerical analysis proved the existence of a recirculating flow structure around the impinging jet. Therefore, understanding the flow field of the impingement jet using nanofluids is crucial. Thus, a detailed numerical simulation is necessary to study the flow field.

Based on the extensive literature survey, the nanofluids-based jet impingement cooling research mainly focused on single jet configuration. The maximum heat transfer enhancement (Nu_nf_/Nu_f_ = 6) is shown in Table 1, obtained by Senkal and Torri [22], who studied multiple jet impingement using Al_2_O_3_-H_2_O nanofluids. Thus, using nanofluids in impinging jet array configuration is promising to be used in thermal management to provide a uniform temperature surface with extremely high heat flux removal. However, the research on multiple jet impingement using nanofluids is scarce.

To the author’s knowledge, the experimental works relevant to nanofluid’s jet array were completed by Senkal and Torri [22,24], Zhou et al. [9], Tie et al. [23], and Darwish et al. [26], and their jet array configuration can be seen in Figure 1. However, the research completed by Zhou et al. [9], Tie et al. [23], and Darwish et al. [26] focused on the effect of nanofluids without considering the impact of jet-to-jet spacing on nanofluids heat transfer in multiple jets; however, they only focused on the single geometrical condition. Although Darwish et al.’s [26] research focused on two different configurations with different nozzle-to-plate heights, they only concentrated on H/d = 10 with constant jet-to-jet spacing, claiming that optimum heat transfer is attained by using water. The research related to jet-to-jet spacing with nanofluids heat transfer in multiple impingement jet cooling is crucial because the interaction between impinging jets might affect the heat transfer of the nanofluid jets array.

Although Senkal and Torri [22,24] investigated the different jet-to-jet spacing of nanofluid’s multiple jet impingement cooling, the number of nozzles is not uniform, refer to Figure 1a,b, so it is hard to give a clear picture of the jet-to-jet effect on nanofluid’s heat transfer in multiple jet flow because of the combined effect of the number of jets. Additionally, sparse research on this topic simultaneously focuses on experimental and numerical work. Senkal and Torri [24] and Darwish et al. [26] investigated nanofluid jets array numerically and experimentally in free jet conditions at H/d of 12.33 and 10, respectively. However, not yet found any experimental and numerical research on this topic has been completed on the submerged jet condition and study of confinement effect at small H/d, which better represents the actual system that exists in electronic cooling practices [5,29].

In addition, the most studied nanofluids are metal oxide-based nanofluids, especially aluminum oxide-based nanofluids that occupied almost half of the research in impingement jet cooling referred to in Table 1 because of their chemical stability and ease of preparation. Recently, Loong et al. [30] have evaluated the heat transfer performance of various metal oxide-water nanofluids in fully developed laminar pipe flow, the evaluated metal oxide included Al_2_O_3_, CuO, MgO, TiO_2_, SiO_2_, ZnO, and ZrO_2_, and the best option is MgO-water nanofluid according to their analysis. Furthermore, the MgO-water nanofluid was found to have excellent features and seemingly can enhance convective heat transfer in low volume (<1 vol%), which is primarily studied in pipe flow cooling and other heat transfer applications, such as car radiator cooling, finned-tube heat exchanger, and corrugated mini channel heat sink [31,32,33,34,35,36]. Thus, this nanofluid is worth studying, but the small amount of research is focused on multiple jets impingement using this type of nanofluid.

The research on nanofluids’ multiple jet impingement cooling is still new and not well assessed. The multiple jet impingement heat transfer is inherently complex because of the high-pressure gradient, flow recirculation, and local thin thermal layer, and it becomes even more complicated when the nanofluid is adopted as a coolant [37,38,39,40,41,42,43]. Furthermore, nanoparticles’ heat transfer enhancement mechanism in base fluid remains unclear [44]. Therefore, the heat transfer characteristics and the hydrodynamics of multiple jet impingement using MgO-H_2_O nanofluids are still not well understood. As a result of all enumerated reasons combined, the flow structure and heat transfer behavior of multiple jet impingement using MgO-water nanofluids with 3 × 3 inline jet arrays was chosen. The jet spacing was set to 3, 4.5, and 6 mm; the Reynold number varies from 1000 to 10,000; and the particle volume fraction ranges from 0% to 0.15%.

## 2. Nanofluids Synthesis and Thermo-Physical Properties

The MgO-water nanofluid is prepared using a two-step method. The two-step method was commonly adopted to prepare nanofluids due to their simplicity, reliability, and economy [18]. MgO-water nanofluids were produced by dispersing an appropriate amount of nanoparticles into distilled water using an ultrasonic bath and mechanical overhead stirrer for 5 h and 1 h. The citric acid was added to enhance the stability of nanofluids, and the sodium hydroxide was used to modulate the pH value of nanofluids for better stability (pH ≈ 10) [45].

The used MgO nanoparticles are bought from Nanostructure and Amorphous Materials, Incorporated, and Table 2 shows the thermal-physical properties of MgO nanoparticles. The morphology of the MgO nanoparticles is spherical and ellipsoidal, and the average particle size is about 20 nm.

The volume fraction percentage of the nanoparticles were chosen to be 0.05%, 0.1%, and 0.15%, respectively. The concentration of nanofluids was deemed diluted, and the nanofluids were observed keeping stable in their stationary state with little or no sediment during experimental runs. The thermal conductivities (k) and viscosity (μ) of the nanofluids are measured by the KD2 pro thermal properties analyzer and by Sine-wave Vibro Viscometer SV-10 at temperature 24 °C, respectively.
(1)$$\rho_{nf}=\phi \rho_{p}+(1-\phi )\rho_{f}$$
(2)$${\mathrm{Cp}}_{\mathrm{nf}}=\frac{\phi \rho_{p}{\mathrm{Cp}}_{p}+(1-\phi {)\rho }_{f}{\mathrm{Cp}}_{f}}{\rho_{\mathrm{nf}}}$$

Furthermore, the other thermal-physical properties of the nanofluid, namely the density (ρ) and heat capacity (Cp), have been calculated from the provided nanoparticles and the pure fluid (i.e., 0 vol%) properties; by Equations (1) and (2). The thermal-physical properties of the nanofluid are given in Table 3.

## 3. Experimental Setup

### 3.1. Test Rig

A test rig was built to investigate the heat transfer characteristic of multiple jet impingements. Figure 2 illustrates the schematic diagram of the test rig for the experimental study and the photograph of the test rig. The coolant is pumped using a submerged brushless motor pump from the cooling tank to the nozzle, then discharged to impinge the hot target surface. The impinged coolant flows back to the cooling tank due to gravity and is recycled again in the system. The coolant recirculates through a cooling block, which is cooled down by a Peltier cooler to maintain the temperature at 24 °C.

The flow rate is monitored using a flow sensor, namely, FS200A G1/2, which is connected to an Arduino, a microcontroller. The sensor sends a signal to the microcontroller, which in turn sends the data to the laptop via USB. The signal is then converted into a value display on the laptop monitor. Additionally, the flow rate of the coolant is regulated by adjusting the power supply to the pump through a voltage controller or by turning a valve.

A 300 W band heater is used to heat the cylindrical aluminum alloy block, and the variac transformer adjusts the heating power. Five thermocouples (TT-K-30-SLE) are used to measure temperature in the respective locations in the test rig, and the milli-volt from thermocouples is measured using a data logger PicoLog TC08 (8 channels). The four thermocouples, T1, T2, T3, and T4, are inserted into the 2 mm diameter straight holes and well terminated at the centerline of the aluminium impingement block, whereas thermocouple T5 is used to measure the temperature of the fluids, which is located in the cooling tank. The measured temperatures from this point are used to compute the heat flux, surface temperature, and average heat transfer coefficient; the details are presented in the next section.

### 3.2. Data Reduction

The average heat transfer coefficient, h_ave_, of the entire target surface is computed by Equation (3), where q is the heat flux on the target surface, T_s_ is the temperature of the target surface, and T_jet_ is the temperature of fluid jets obtained from temperature data of thermocouple T5.
(3)$$h_{\mathrm{ave}}=\frac{q}{T_{s}-T_{\mathrm{jet}}}$$

By fitting the temperature data from thermocouples T1, T2, T3, and T4 using the linear regression method, Equation (4) is formed, taking the impingement surface as the origin and the positive z-coordinate is downward along the centerline of the cylindrical block. Two information can be extracted from Equation (4): temperature gradient (dT/dz), which is estimated through the best-fit approach, and the surface temperature (T_s_), which functions as the intercept of T(z).
(4)$$T(z)=\left({\left.\frac{dT}{dz}\right|}_{\mathrm{best}\;\mathrm{fit}}\right)z+T_{s}$$

Assuming that the measurement has reached a steady state, and that the Teflon insulation layer prevents heat leak in the radial direction, it is reasonable to assume that the heat flux along the block is uniform. Hence, Equation (5) can be used to calculate the heat flux on the target surface.
(5)$$q=k_{\mathrm{Al}}\frac{dT}{dz}$$

The dT/dz is the temperature gradient along the z-coordinate in Equation (4), and the k_Al_ is the thermal conductivity of the heater block, which is made of aluminum alloy (wrought 6061), and it is temperature dependent. Thus, Equation (6), which was obtained by regression of the available data from [47], is used to compute the thermal conductivity of the heater block under average block temperature, T_ave_. The T_ave_ is the average temperature obtained from T1, T2, T3, and T4. The temperatures T1 to T4 showed a linear variation, but, since they are closely located, this variation is considered small. Therefore, the average temperature can represent the temperature of the block.
(6)$$k_{\mathrm{Al}}=0.003495T_{\mathrm{ave}}{}^{2}+0.103652T_{\mathrm{ave}}+191.9451$$

Consequently, the surface average Nusselt number is computed based on Equation (7), where have is the heat transfer coefficient obtained from Equation (3), the D is the diameter of the impingement hot surface, and the k_nf_ is the thermal conductivity of nanofluids.
(7)$$Nu_{D}=\frac{h_{\mathrm{ave}}D}{k_{\mathrm{nf}}}$$

### 3.3. Uncertainty Analysis

In an experimental study, uncertainty analysis is crucial in determining the uncertainty of experimental data parameters, such as the heat transfer coefficient, Reynolds number, and Nusselt number. This procedure is necessary to verify the experimental measurement, and it also helps to reduce experimental error. However, many independent variables, such as flow rate and temperature measurement, affected the experimental data parameters. In this regard, Kline and McClintock’s [48] method was applied for experimental data uncertainty. The uncertainty of each experimental data parameter, U*_f_*, is calculated by Equation (8).
(8)$$U_{f}=\sqrt{{\sum_{i=1}^{n}\left(\frac{\partial f}{\partial w_{i}}U_{w_{i}}\right)}^{2}}$$
where U*_w_*_i_ is the uncertainty of directly measured quantity *w*_i_, and *f* is the dependent variable. The 95% probability level is applied to fulfill scientific research requirements. The maximum uncertainty measurements were ±3.1% for heat transfer coefficient, ±2.84% for Reynolds number, and ±6.1% for Nusselt number. Table 4 lists the uncertainty of the directly measured quantity.

### 3.4. Validation of Measurement

Since there are no existing published results of multiple impingement jets that match the experimental setup described in this paper, it is not possible to make any comparisons. Therefore, to validate the experimental results, the Nusselt number obtained from a single impingement jet using water at H/d ratio of 1 was compared with the correlation equation proposed by Steven and Webb [49] in Equation (9). The comparison is shown in Figure 3, which indicates that the experimental data falls within the range of correlation data with a maximum deviation of ±15%. As a result, it can be concluded that the experimental data is consistent with the correlation data, thereby demonstrating the reliability of the results obtained from the current test rig.
(9)Nu=3.62Red0.35(H/d)−0.032Pr0.41+2(D/2d)21.48e−0.56(D/2d)−0.56D/2d−1−0.56+1.48(−0.56)2−7−1/7

## 4. Numerical Analysis

### 4.1. Problem Description

Numerical simulation is an essential tool to investigate the flow field of multiple jet impingement that is challenging to capture in the experiment. Figure 4 illustrates the present study’s physical condition of multiple jet impingement. The selection of the computational domain for the numerical simulation is depicted in Figure 5. The multiple jet impingements with nozzle diameter of 1.5 mm, jet-to-jet spacing, s = 3.0, 4.5, and 6.0 mm, respectively, and nozzle-to-plate distance, H = 3 mm, are investigated numerically.

Since the flow is symmetric to the XZ-plane (symmetric plane 1) and YZ-plane (symmetric plane 2), only a quarter of the domain is considered in this numerical analysis. Thus, the symmetric boundary condition is applied to these two planes to reduce the entire model into a quarter model, as shown in Figure 6. Then, this domain is discretized into a mesh model suitable for numerical evaluation.

Teamah et al. [14], Senkal and Torri [24], and Darwish et al. [26] have numerically investigated jet impingement using nanofluids with a high-volume fraction (>1 vol%) based on experimental studies. They treated the nanofluid as a Newtonian fluid and adopted a single-phase model that assumes constant thermal and physical properties for the simulation of nanofluid jet flow. Given that a small volume fraction (<1 vol%) was used in the present study, it is feasible to utilize the same approach for simulating nanofluids in multiple jet impingement. Therefore, the following assumptions were made for the current numerical analysis: the nanofluid is considered an incompressible, Newtonian fluid in which the mixture of nanoparticles and base fluids is treated as one phase. The multiple jet flow is assumed to be steady turbulent. The effective thermal and physical properties of the nanofluids (as listed in Table 2) are considered constant and are used as material properties for the simulation.

The use of relatively low heat flux in the present experimental process effectively prevents boiling and negligible buoyancy effects. The absence of a significant temperature gradient and substantial differences in fluid densities precludes the emergence of buoyancy effects. Therefore, the influence of buoyancy is insignificant.

### 4.2. Mathematical Formulation

#### 4.2.1. Governing Equations

After the multiple jet impingement flow reaches steady, the space between the nozzle plate and impingement surface is filled with liquid due to the confinement effect. The jet array is in submerged condition, and the flow pattern is hard to observe by the naked eye; hence ANSYS FLUENT 19.2 software is used to solve the conservation equations for mass, momentum, and energy to simulate the flow field and heat transfer behavior of multiple jets impingement. The detail of the governing equations can be found in the ANSYS Fluent theory guide [50]. The relevant equations are presented as follows:

Continuity equation
(10)$$\frac{\partial \rho }{\partial t}+\nabla •\left(\rho \vec{V}\right)=0$$

Momentum equation
(11)$$\frac{\partial }{\partial t}(\rho \vec{V})+\nabla •\left(\rho \vec{V}\vec{V}\right)=-\nabla p+\nabla \tau +\rho g$$

Energy equation
(12)$$\frac{\partial }{\partial t}(\rho E)+\nabla •\left(\vec{V}(\rho E+p)\right)=\nabla •\left(k\nabla T+\tau \vec{V}\right)$$

Here, ρ is the density of the fluid, p is the static pressure, $\vec{V}$ is velocity vector, **τ** is the stress tensor, g is the gravitational acceleration, k is the thermal conductivity of the fluid, E is the total energy, T is the temperature of fluids, and t is the time.

#### 4.2.2. Turbulent Model

The modeled transport equations for *k* and ω of SST *k*-ω turbulent model are shown as Equations (13) and (14). G*_k_* represents the generation of turbulence kinetic energy due to the mean velocity gradients in these equations. G_ω_ is the generation of ω. Γ*_k_* and Γ_ω_ represent the effective diffusivity of *k* and ω, respectively. Y*_k_* and Y_ω_ represent the dissipation of k and ω due to turbulence. D_ω_ represents the cross-diffusion term. S*_k_* and S_ω_ are user-defined source terms.
(13)$$\frac{\partial }{\partial t}(\rho k)+\frac{\partial }{\partial x_{i}}(\rho ku_{i})=\frac{\partial }{\partial x_{j}}\left[\Gamma_{k}\frac{\partial k}{\partial x_{j}}\right]+G_{k}-Y_{k}+S_{k}$$
(14)$$\frac{\partial }{\partial t}(\rho \omega )+\frac{\partial }{\partial x_{j}}(\rho \omega u_{j})=\frac{\partial }{\partial x_{j}}\left[\Gamma_{\omega }\frac{\partial \omega }{\partial x_{j}}\right]+G_{\omega }-Y_{\omega }+D_{\omega }+S_{\omega }$$

The effective diffusivities for the SST *k*-ω turbulent model are given by Equations (15) and (16):(15)$$\Gamma_{k}=\mu +\frac{\mu_{t}}{\sigma_{k}}$$
(16)$$\Gamma_{\omega }=\mu +\frac{\mu_{t}}{\sigma_{\omega }}$$

Moreover, μ_t_ is the turbulent viscosity, which is defined as follows:(17)$$\mu_{t}=\frac{\rho k}{\omega }\frac{1}{\max\left[\frac{1}{\alpha^{*}},\frac{SF_{2}}{\alpha_{1}\omega }\right]}$$
where $\sigma_{k}$ and $\sigma_{\omega }$ are the turbulent Prandtl number for *k* and ω, α*, and α_1_ are the model coefficient and model constant, F_2_ is the blending function, and S is the modulus of the mean rate-of-strain tensor, which is given by:(18)$$S=\sqrt{\frac{1}{2}\left(\frac{\partial u_{j}}{\partial u_{i}}+\frac{\partial u_{i}}{\partial u_{j}}\right)\left(\frac{\partial u_{j}}{\partial u_{i}}+\frac{\partial u_{i}}{\partial u_{j}}\right)}$$

### 4.3. Boundary Conditions

Uniform inlet velocity and temperature are imposed on the nozzle inlet. The jet velocity is according to the Reynolds number (1000–10,000), and the inlet temperature is consistent with the experiment (24 °C). The opening of the fluid domain is set to an entrainment outlet with zero static pressure in order to let the fluid flow in and out freely depending on the flow condition. Symmetric boundary conditions are applied to the ZX and ZY planes. A constant heat flux of 30,000 W/m^2^ is applied to the target surface, and other surfaces are regarded as adiabatic. Non-slip condition is set to all wall surfaces. The distribution of turbulent kinetic energy, *k*, at inlet and outlet is determined by:(19)$$k=\frac{3}{2}{\left(u_{\mathrm{ave}}I\right)}^{2}$$

The distribution of dissipation rate *k* at the inlet is determined by:(20)$$\omega =\frac{k^{1/2}}{C_{\mu }^{1/4}d_{h}}$$

The distribution of dissipation rate ω at the outlet is determined by:(21)$$\omega =\rho \frac{k}{\mu }{\left(\frac{\mu_{t}}{\mu }\right)}^{{}^{-1}}$$
where I is turbulent intensity, and the default value of 5% is used in this case. C_μ_ is the empirical constant specified in the turbulent model with an approximate value of 0.09. d_h_ is the hydraulic diameter, and nozzle diameter is considered in this case. μ_t_/μ is the turbulent viscosity ratio, and the default value 10 is used in this case.

### 4.4. Meshing Model

A three-dimensional quarter model represents the fluid domain of multiple jet impingement in the present study. The quarter model is considered because the flow field is assumed symmetrical to planes XZ and YZ, and the computation cost is reduced at least four times more than a full model. The combination of hexahedral meshes and prisms meshes was used for the computation. The number of hexahedral meshes is more than prism meshes to minimize the numerical diffusion and get a better solution. The mesh is arranged layer by layer clustered toward the walls in the z-direction; a trial-and-error process is conducted to make sure the mesh in the near-wall region is fined enough so that the height of the first cell adjacent to the wall falls in the desired y+ range to resolve the viscous sub-layer, which is y + ~1 for SST *k*-ω turbulent model. Furthermore, the mesh is refined in the impinging jet zone because large flow fluctuations are expected in this region. The mesh quality of the present model is considered good because the skewness and orthogonal quality of the present mesh are below 0.9 and above 0.1, respectively, according to the Fluent user manual. The meshing models of multiple jet impingement of s/d = 4, 3, and 2, at H/d = 2, respectively, are depicted in Figure 7.

### 4.5. Numerical Solution

After integrating the governing equations over finite volumes, a set of linear algebraic equations is generated. To solve the solution between the cell’s centroid and cell linearly, the least-squares cell-based gradient method is employed for spatial discretization. The pressure-based solver uses the PRESTO interpolation scheme. The convection or diffusion terms in the momentum equation and the ω from k-ω turbulent models are discretized using the first-order upwind interpolation scheme. On the other hand, the second-order upwind interpolation scheme is adopted to discretize other convection or diffusion terms, such as the k from the SST k-ω turbulent model and the energy term. While the second-order upwind interpolation scheme can provide more accurate results, the first-order upwind interpolation scheme is applied as mentioned earlier to ensure better convergence. It should be noted that the numerical results have been validated using experimental data.

The convergence criteria of 10^−6^, 10^−4^, 10^−4^, and 10^−5^ are set for the residuals of energy, *k*, ω, and other variables, respectively, in this numerical study. When all residual variables are declined below the given residuals value, the numerical calculation is stopped. Moreover, the average heat transfer coefficient of the target surface is also monitored with the convergence criteria of 10^−5^.

### 4.6. Grid Independence Study and Grid Convergence Index Evaluation

A grid independence study is performed to verify a numerical model, which involves gradually refining mesh size from a very coarse grid to a very fine grid until constant results are obtained so that the result is considered independent from the mesh size. Thus, finding the optimum mesh size is important to save more computation power and time. Four grid levels with different element numbers are considered for each model: very coarse, coarse, fine, and very fine. For the s/d = 4 model, the evaluated element numbers are 72,071, 157,114, 385,543, and 782,680, respectively. For the s/d = 3 model, the evaluated element numbers are 71,761, 156,410, 388,250, and 771,654, respectively. For the s/d = 2 model, the evaluated element numbers are 71,575, 159,450, 398,042, and 777,894, respectively.

Table 5 summarizes the grid independence and convergence index results for all models under jet speed of 5.31 m/s using water as coolant. By considering the deviation percentage, the fine grid of all models is adopted in the present numerical study since the numerical solution is proved to be independent of grid size as the Nusselt number result deviation between fine and very fine grids falls below 3% for all models.

Furthermore, the verification process of numerical study also involves error estimation by putting an error band on the computed value, which is obtained by calculating the Grid Convergence Index (GCI). The GCI for the fine grid models adopted in this study is estimated based on the method proposed by Roache [51]. The maximum GCI of the fine grid models is 4.6%, which is less than 5% for all models. Thus, it is feasible to select fine grid models for further calculation since they achieved grid independence with low GCI.

### 4.7. Model Validation

A validated model means a model has a high degree of representing the real world accurately. Water is considered an appropriate coolant for model validation because water is a commonly studied coolant in both experimental and numerical studies, and it is used as the base fluid to prepare nanofluids. Thus, a comparison between numerical and experimental results for multiple jet impingement with s/d = 4, 3, and 2 using water as the coolant is made to validate the model, which is illustrated in Figure 8. The average deviation between numerical and experimental results is about 3.75%, which falls within the maximum GCI ±4.6%. Thus, the present model is in good agreement with the experimental data, so these results ensure that the current model can be used to investigate the flow field of the multiple jet impingement in this paper.

## 5. Results and Discussion

### 5.1. Heat Transfer Prediction on Multiple Jet Impingement Using Single Phase Model

Figure 9 illustrates the experimental and numerical Nusselt number versus Reynolds number results of multiple jet impingement with various s/d dimension ratios using nanofluids with 0, 0.05, 0.1, and 0.15 vol%, respectively. As observed, the numerical result using the single-phase model is in good agreement with measured data by using water (i.e., 0 vol %) as working fluid since the average deviation between numerical and experimental results is about 3.75%, which falls within the maximum GCI 4.6%. In contrast, the numerical result can predict the trend of the heat transfer behavior of nanofluid’s multiple jet impingement correctly, but with pronouncing deviation from experimental results, especially at a regime of high Reynolds number, with an average deviation from experimental data of 7.9%, 11.2%, and 6.3% for numerical data using nanofluids of 0.05, 0.1, and 0.15 vol%, respectively, which is overall larger than the maximum GCI 4.6%. Thus, this concludes that the single-phase model fails to capture the effect of nanoparticles in the flow field since it is only considered the effective property of the fluid.

### 5.2. Fluid Flow Structure and Heat Transfer

Logically, the Nusselt number increases with increasing Reynolds number due to the more substantial convective effect. However, to explain the impact of dimension ratio s/d on heat transfer, the flow fields of multiple jet impingement with different jet-to-jet spacing need to be investigated. Figure 10 depicts the overall views, side view, and focused view between impinging jets of flow fields for multiple jet impingement with dimension ratio s/d of 4, 3, and 2. For brevity, the flow fields of multiple jet impingement using water at the jet speed of 5.3 m/s are presented only in this paper because the multiple jet impingement from the simulation with other jet speeds and volume fractions show a similar flow pattern.

Only a quarter view of the flow field is presented because the flow field is symmetrical to ZX and ZY, and the flow field is represented by 3D streamlines shown in Figure 10. The fluid jet issues from the nozzle hit the target surface and formed a radial flow. The radial flows interact midway between impinging jets reflect from the target surface to the nozzle plate generating a recirculating flow structure. Then, the fluid leaves the impinging zone in a screw-like pattern in the X and Y direction. Furthermore, some vortical flow structure exists between the screw-like flow and the fluid flow in the diagonal direction outside the impinging jet zone.

The flow field patterns of multiple jet impingement are distinct for different dimension ratios s/d. The recirculating flow structure becomes smaller as the jet-to-jet spacing decreases, and the size of the screw-like flow structure leaving the impinging jet zone also becomes smaller. As a result, the fluid exits the impinging jet zone faster with a smaller jet-to-jet spacing. Furthermore, the fluid flow in a diagonal direction outside the impinging jet zone is flattened as the jet-to-jet spacing of impinging jet decreases, and consequently, a secondary recirculating flow structure is generated at a dimension ratio of s/d = 2.

The Prandtl number is a significant factor in understanding how heat moves between fluid jets and the surrounding fluid. It is a dimensionless parameter that represents the ratio of momentum diffusivity to thermal diffusivity in a fluid. When the Prandtl number is high, like in nanofluids or water, the heat transfer rate is more sensitive to the convective effect of the jets flow. This causes the heat to concentrate more around where the jets hit the fluid, resulting in a more concentrated heat transfer distribution.

However, the specific effect of the Prandtl number on heat transfer distribution in multiple jet systems also depends on the jet spacing. The changing flow pattern with decreasing jet-to-jet spacing affects the heat transfer behavior of multiple jet impingement. Figure 11 shows the temperature distribution on the target surface for multiple jet impingement with a dimension ratio s/d of 4, 3, and 2. The overall temperature of the target surface is colder at a higher jet speed (u_jet_ = 0.7 m/s) compared to a lower jet speed (u_jet_ = 5.3 m/s) due to the high convective effect. The lowest temperature can be achieved at the stagnation zones in the impinging jets zone since the boundary layer is thin in this region due to directly striking by fluid jets. The effective area of the impinging jet zone with low temperature becomes smaller as the dimension ratio s/d decreases, resulting in a heat transfer degradation.

On the other hand, the temperature at the radial flow interaction region is higher than in stagnation zones of impinging jet because the fluid flow is slowed down due to the collision of radial flow from the adjacent jets. Furthermore, a significant temperature escalation is found in the peripheral region of the target surface in the diagonal direction, and the temperature in this region is even higher for multiple jet impingement with a smaller jet-to-jet spacing. As the dimension ratio s/d decreases, the fluid flow in the diagonal direction outside the impinging jet zone loses its momentum because of the enhancement of wall friction due to the increasing length of the wall jet zone. Thus, when the dimension ratio s/d = 2, the wall jet in the enumerated direction entrains back and forms a secondary recirculating flow structure shown in Figure 10; Consequently, the stagnant fluid after the secondary flow structure causes drastic temperature enhancement at the target surface’s edge.

In short, the reduction in effective area of the impinging jet zone and the degradation of heat transfer at the edge of the target surface when the dimension ratio s/d decreases explain the reduction trend of Nusselt number at a smaller dimension ratio s/d under the same Reynolds number for water and all nanofluids (0.05, 0.1, and 0.15 vol%, respectively).

### 5.3. Nusselt Number Enhancement Ratio Results

In order to better understand the heat transfer behavior of multiple jet impingement using nanofluids as working fluid, the Nusselt number enhancement ratio (Nu_nf_/Nu_f_) is evaluated under different Reynolds numbers, dimension ratio s/d, and nanoparticle concentrations. The nanofluid enhances heat transfer of multiple jet impingements if the value of Nu_nf_/Nu_f_ is greater than unity. Furthermore, the Nusselt number enhancement ratio of nanofluid is relative to water’s Nusselt number; thus, it can reflect the effect of the suspended nanoparticles in base fluid that cannot capture using a single-phase model in numerical simulation on heat transfer of multiple jet impingement using nanofluids.

Figure 12 depicts the variation of the Nusselt number enhancement ratio of MgO-water nanofluid versus jet-to-jet spacing with different Reynolds numbers at varying particle volume fractions. According to Figure 12, the Nusselt number enhancement ratio of nanofluid’s multiple jet impingement is decreased with increasing Reynolds number and attained a value lower than unity for some cases. Thus, this indicates that using nanofluids for multiple jet impingement at a high Reynolds number regime is not beneficial.

Furthermore, the Nu_nf_/Nu_f_ generally decreases with increasing dimension ratio s/d, and the trend tendency is more apparent at a lower Reynolds number. Thus, using a large jet-to-jet spacing for a nanofluid’s multiple jet impingement is not beneficial since it can be observed that the measured Nu_nf_/Nu_f_ for all cases in current experimental work with the largest jet-to-jet spacing, s/d = 4, is always lower than unity, which indicates degradation of heat transfer.

Moreover, using a higher volume fraction of multiple jet impingement can enhance heat transfer of multiple jet impingement. For instance, an optimum Nu_nf_/Nu_f_ with about 1.15 was discovered using a 0.15 vol% nanofluid, which is the highest volume fraction used in current experimental work.

Thus, in short, the MgO-water nanofluid can provide a heat transfer enhancement at a small jet-to-jet spacing using a high particle volume fraction under a low Reynolds number; otherwise, an adverse effect on heat transfer may occur. Within the range of experimental parameters considered, it has been discovered that the optimum Nusselt number enhancement ratio can be achieved using an intermediate dimension ratio s/d of 3 using the highest nanoparticle volume fraction of 0.15 vol% under the lowest Reynolds number of 1013.

### 5.4. Explanations on Heat Transfer Behavior of Multiple Jet Impingement Using Nanofluids

Regarding the effect of suspended nanoparticles in base fluid on the Nusselt number enhancement ratio relative to pure water, the experimental results revealed that the nanofluids do not necessarily provide heat transfer enhancement; instead, a degradation of heat transfer is observed mainly at high Reynolds number under large dimension ratio s/d.

Furthermore, Nguyen et al. [4] and Senkal and Torri [23] have also discovered that the nanofluid gave the worst cooling performance than pure water under nearly the same Reynolds number for some cases in single and multiple jet impingement, respectively. They explained that the degradation of heat transfer using nanofluids is caused by recirculating flow structure in the jet flow. This recirculating flow structure entraps the nanofluid, which has a higher thermal conductivity than pure water. Thus, the trapped fluid in this region gets warm faster, and the thermal boundary layer reaches its liquid surface resulting in heat transfer reduction on the impingement surface.

The numerical simulation in the present paper proves the existence of recirculation and vortical flow structure in the multiple jet impingement; see again Figure 10. Therefore, the description of Nguyen et al. [4] and Senkal and Torri [23] on nanofluid heat transfer is reasonable and acceptable. However, they did not include the effect of Reynolds number and jet-to-jet spacing. In this regard, the present paper further extends their explanation by having the mentioned effects in order to explain the heat transfer behavior of multiple jet impingement using nanofluid.

The recirculating and vortical flow structure in the multiple jet impingement traps the nanoparticles and promotes a higher collision rate between the particles as the velocity of the flow field increases. The high collision rate of nanoparticles increases the chance of nanoparticle aggregates. The aggregation of nanoparticles drastically enhances the viscosity of fluid that negatively affects the heat transfer of multiple jet impingement due to the reduction in convective effect. The velocity of the flow field increases when the jet speed increases, indicating the Reynolds number’s rise; thus, this explains the decreasing trend of the Nusselt number enhancement ratio with the growth of the Reynolds number.

The heat transfer of multiple jet impingement is enhanced using a high nanoparticle volume fraction (0.15 vol%) because the increasing volume fraction of nanoparticles also indicates a rising number of particles in the multiple jet flow, which boost the thermal dispersion effect of nanoparticles and accelerate the energy exchange in the fluid. However, the benefit of using nanofluid depends on geometrical configuration since the worst heat transfer performance is found for multiple jet impingement with a large dimension ratio s/d.

The heat transfer of multiple jet impingements using nanofluid is generally degraded as the dimension ratio s/d increases under the same Reynolds number. The recirculating flow structure grows with increasing jet-to-jet spacing, see Figure 10; thus, this prolongs the resident time of nanofluids in the impinging jet zone. The entrapped nanofluid heats faster due to its high thermal conductivity resulting in a thick thermal boundary layer on the impingement surface that causes heat transfer degradation relative to water. In other words, when the jet-to-jet spacing decreases, the entrapped hot nanofluids can leave quickly from the impinging jet, and thus the heat transfer performance is generally better than multiple jets with a large dimension ratio s/d.

## 6. Conclusions

An experimental and numerical investigation using MgO-water nanofluids as working fluids for the multiple jet impingement is performed in the range of 0 ≤ ϕ ≤ 0.15, 2 ≤ s/d ≤ 4, and 1000 < Red < 10,000 at H/d = 2 to study the flow structure and heat transfer behavior. The present paper is summarized as follows:

The study shows that using a single-phase model to predict the heat transfer performance of nanofluid multiple jet impingement is limited. While the model can accurately predict the behavior of water, it deviates significantly from experimental results when nanofluids are used, particularly at high Reynolds numbers. This indicates that the model is unable to fully capture the effect of nanoparticles in the flow field. Thus, the author recommends that using multiple phase models can better account for the complex interaction between nanoparticles and the base fluid in multiple jet impingement flow.The flow pattern of multiple jet impingement is greatly affected by changes in the geometry, specifically the jet-to-jet spacing. The interaction between radial flows from adjacent jets generates recirculating flow structures, which become smaller as the jet-to-jet spacing decreases. At a dimension ratio of s/d = 2, the loss of momentum in the diagonal flow outside the impinging jet zone causes the formation of a secondary recirculating flow structure, leading to a significant temperature escalation at the target surface’s edge. The screw-like flow structure leaving the impinging jet zone also becomes smaller as the jet-to-jet spacing decreases, resulting in a faster exit of the fluid from the impinging jet zone. These changes in flow pattern have a significant impact on heat transfer behavior, leading to a reduction in the Nusselt number at smaller dimension ratios s/d under the same Reynolds number for water and all nanofluids (0.05, 0.1, and 0.15 vol%).The study evaluated the heat transfer behavior of multiple jet impingement using MgO-water nanofluid by analyzing the Nusselt number enhancement ratio (Nu_nf_/Nu_f_) under different Reynolds numbers, dimension ratio s/d, and nanoparticle concentrations. The results show that using nanofluids for multiple jet impingement at a high Reynolds number regime and using a large jet-to-jet spacing is not beneficial. An optimum Nu_nf_/Nu_f_ of 1.15 was achieved using a 0.15 vol% nanofluid with an intermediate dimension ratio s/d of 3 under the lowest Reynolds number of 1013. The study concludes that the nanofluid can enhance heat transfer at a small jet-to-jet spacing using a high particle volume fraction under a low Reynolds number.The present paper used numerical simulation to confirm the existence of recirculation and vortical flow structures in multiple jet impingement, which supports the previous descriptions by Nguyen et al. [4] and Senkal and Torri [23] on nanofluid heat transfer. However, those previous studies did not consider the effects of Reynolds number and jet-to-jet spacing, which the present paper addressed to further explain the heat transfer behavior of multiple jet impingement using nanofluid.The recirculating and vortical flow structure in multiple jet impingement increases the collision rate of nanoparticles and promotes nanoparticle aggregation, reducing convective heat transfer and decreasing the Nusselt number enhancement ratio as Reynolds number increases. Using a high nanoparticle volume fraction enhances thermal dispersion and accelerates energy exchange, but the performance depends on geometrical configuration. Multiple jet impingement with a large dimension ratio has the worst heat transfer performance due to a thick thermal boundary layer caused by the entrapped nanofluid heating faster, while decreasing jet-to-jet spacing allows entrapped hot nanofluids to leave quickly and improves heat transfer performance.

In conclusion, this study provides valuable insights into the heat transfer behavior of MgO-water nanofluids under multiple jet impingement. The findings emphasize the significance of the interplay between nanoparticles and flow patterns in heat transfer enhancement. The results of this study have practical implications for optimizing the cooling performance of multiple jet impingement using MgO-water nanofluids, which can benefit various industrial applications.

## Figures and Tables

**Figure 1 materials-16-03942-f001:**
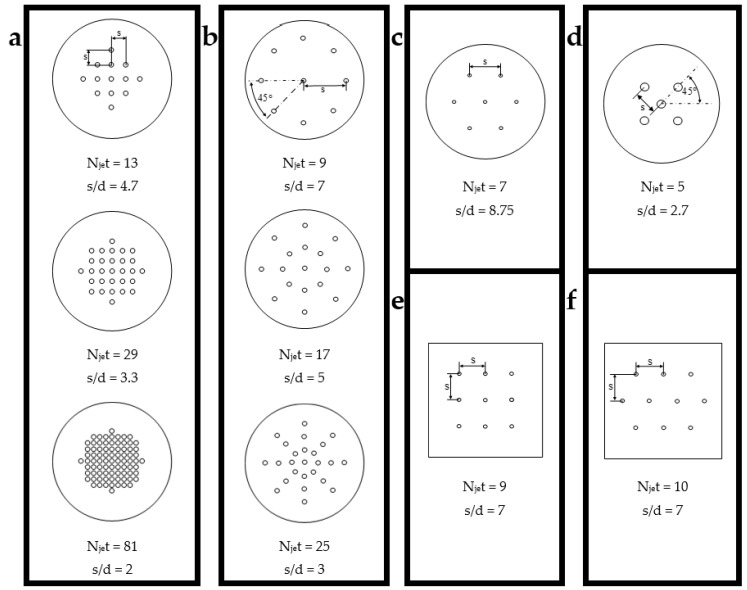
Arrangement of nozzles studied by (**a**) Senkal and Torri [22]; (**b**) Senkal and Torri [24]; (**c**) Zhou et al. [9]; (**d**) Tie et al. [20]; (**e**) Darwish et al. [26] (inline); (**f**) Darwish et al. [26] (staggered).

**Figure 2 materials-16-03942-f002:**
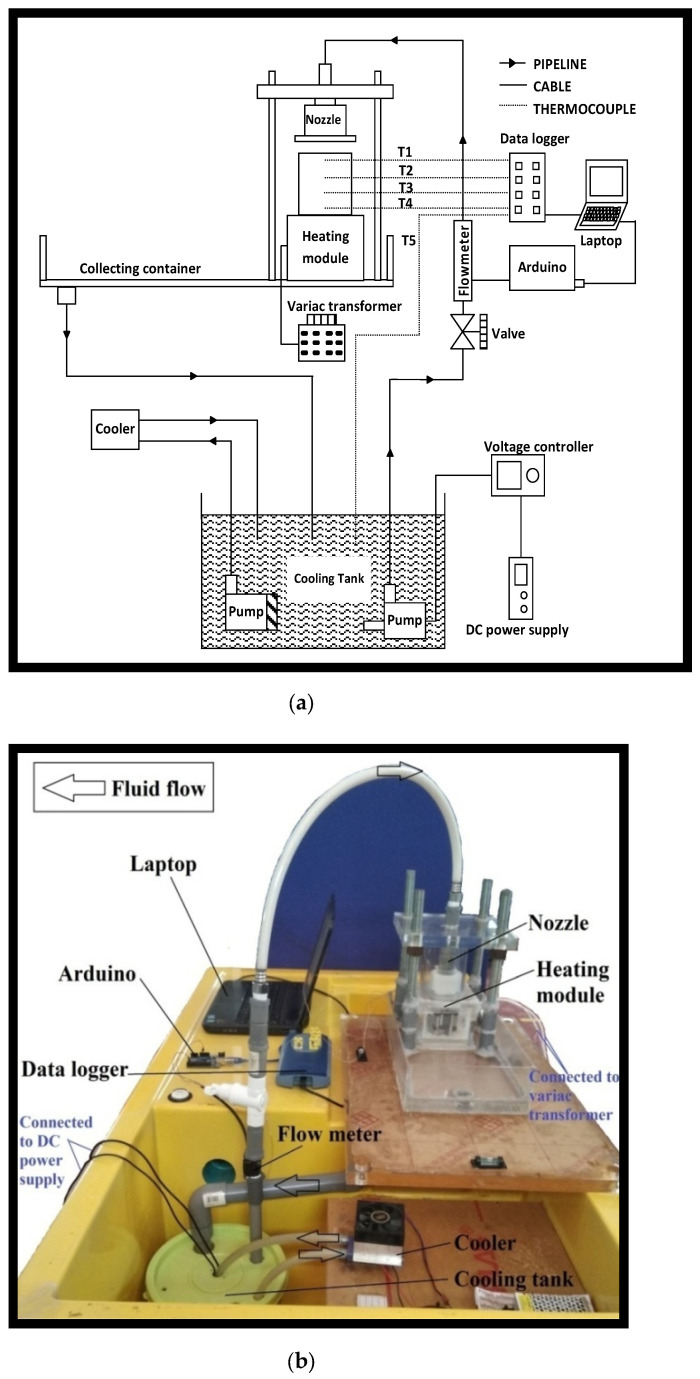
(**a**) Schematic Diagram of the Test Rig. (**b**) Test rig photograph.

**Figure 3 materials-16-03942-f003:**
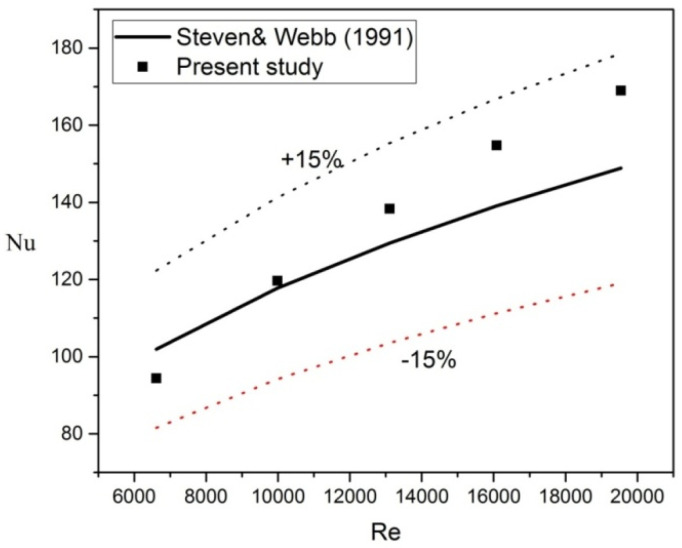
Average Nu vs. Re for Single Impingement Jet using Water at H/d = 1 [49].

**Figure 4 materials-16-03942-f004:**
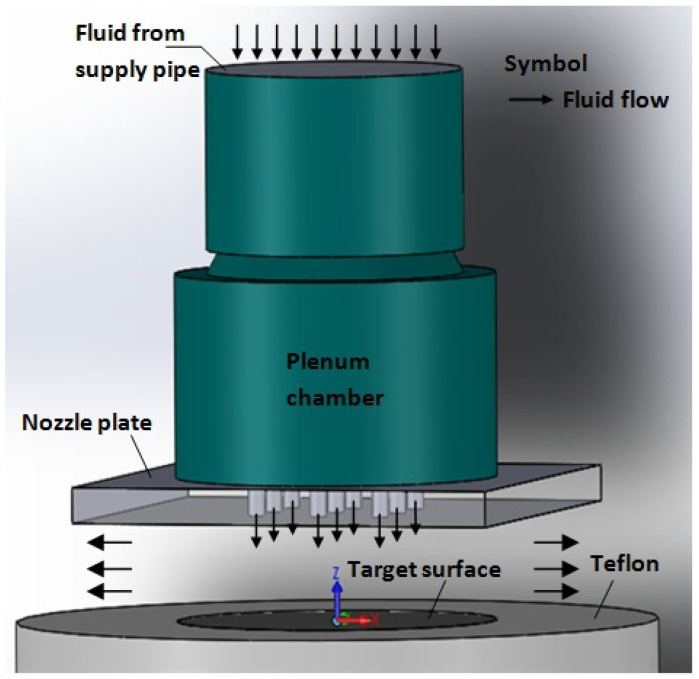
Physical Condition of Multiple Jet Impingement.

**Figure 5 materials-16-03942-f005:**
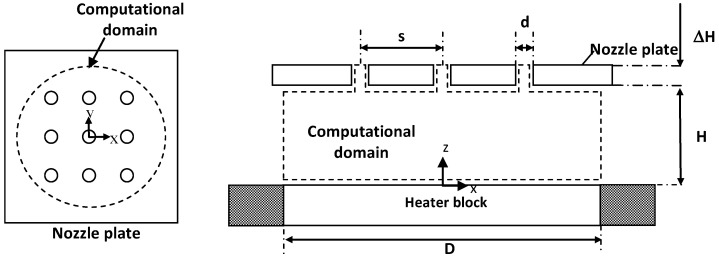
Top View and Side View of the Selected Computational Domain for Numerical Simulation.

**Figure 6 materials-16-03942-f006:**
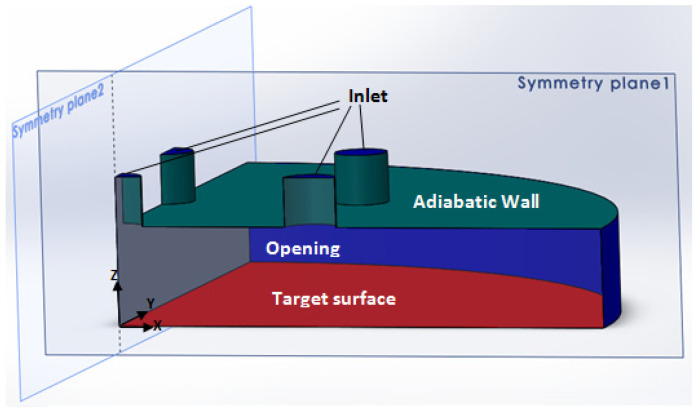
Quarter Part of the Computational Domain with Naming of Boundary Condition.

**Figure 7 materials-16-03942-f007:**
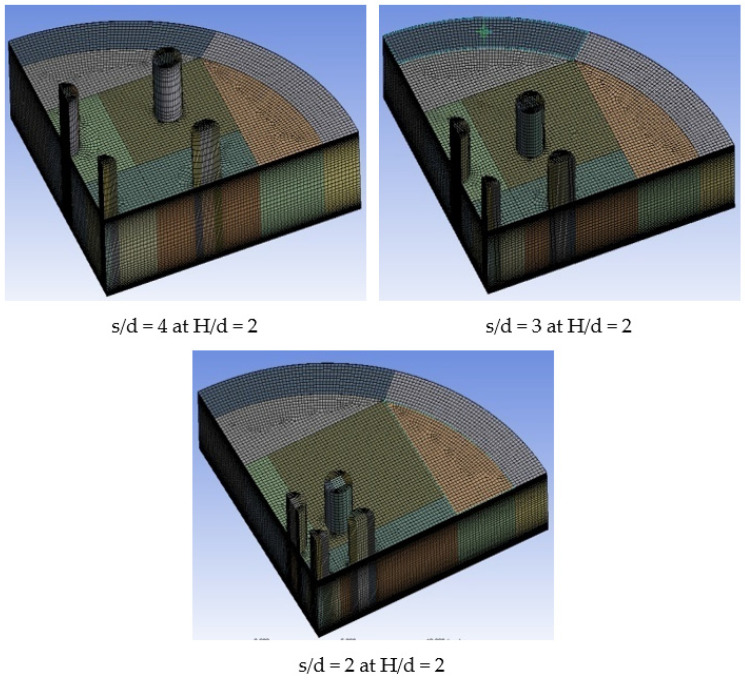
The meshing models for s/d = 4, 3, and 2 at H/d = 2.

**Figure 8 materials-16-03942-f008:**
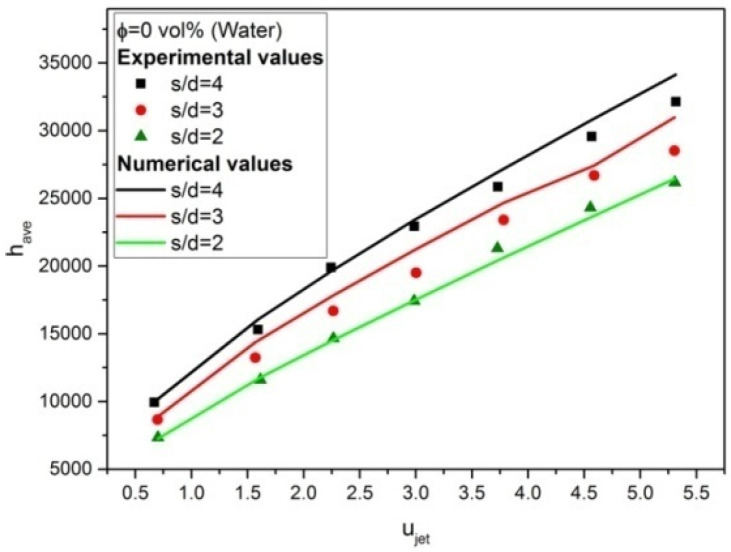
Comparison between experimental and numerical results for multiple jet impingement with s/d = 4, 3, and 2 using water as coolant.

**Figure 9 materials-16-03942-f009:**
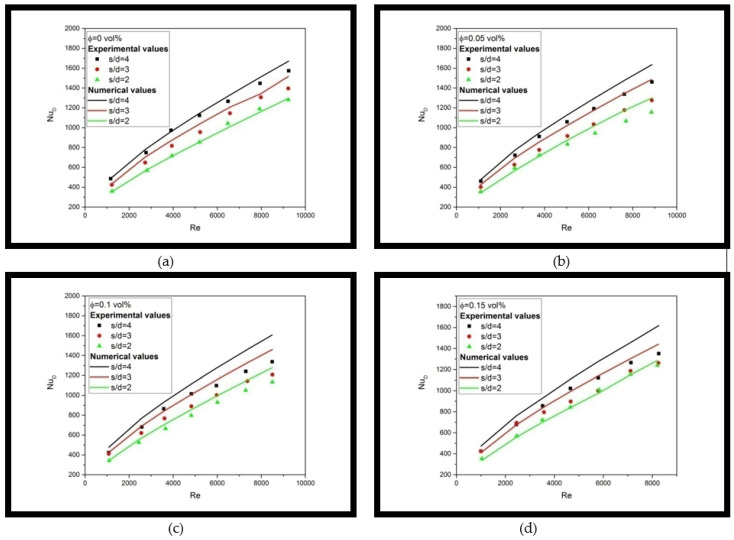
Comparison between Experimental and Numerical Results for Multiple Jet Impingement with s/d = 4, 3, and 2 using Nanofluids with (**a**) 0 vol% (**b**) 0.05 vol% (**c**) 0.1 vol% (**d**) 0.15 vol%.

**Figure 10 materials-16-03942-f010:**
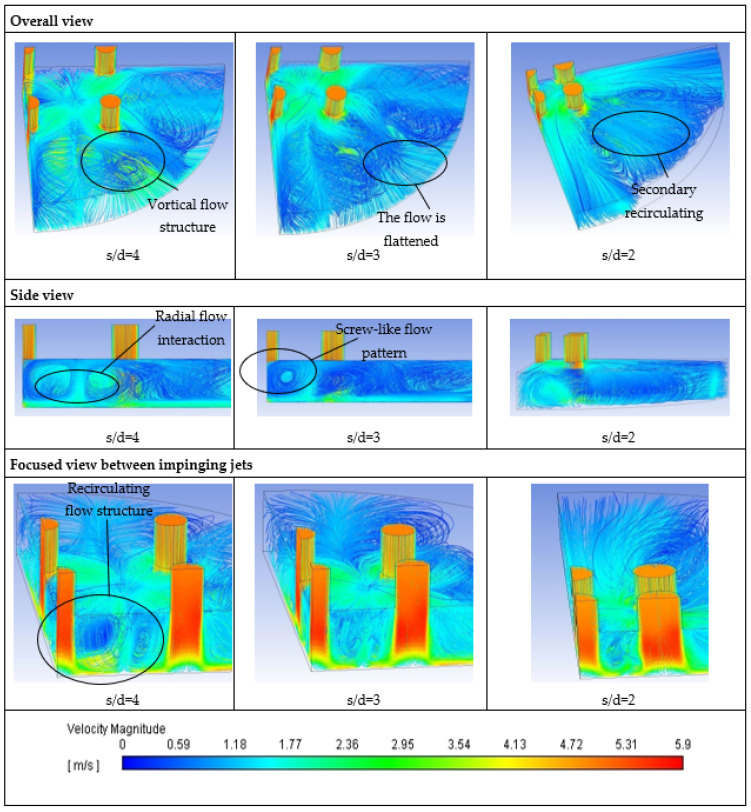
Overall View, Side View, and Focused View between Impinging Jets of Flow Fields for Multiple Jet Impingement s/d = 4, s/d = 3, and s/d = 2.

**Figure 11 materials-16-03942-f011:**
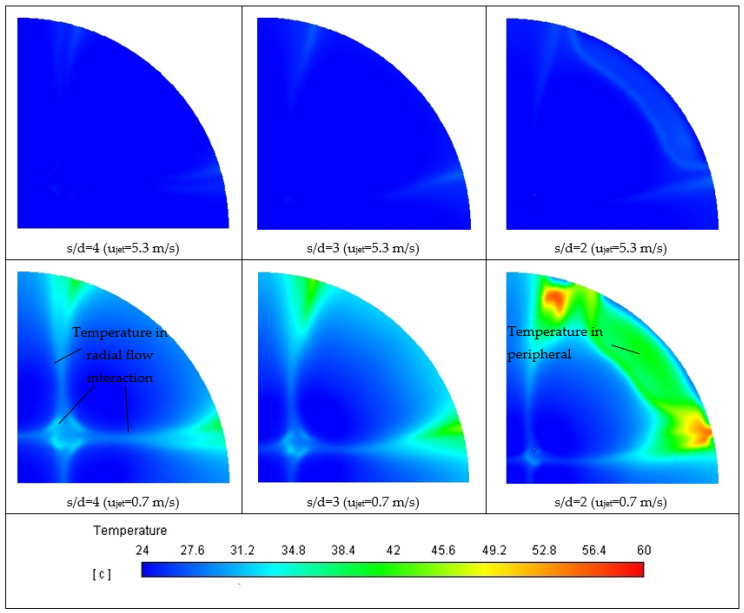
Distribution Temperature of Target Surface for Multiple Jet Impingement s/d = 4, s/d = 3, and s/d = 2 with Jet Speeds of 5.3 m/s and 0.7 m/s.

**Figure 12 materials-16-03942-f012:**
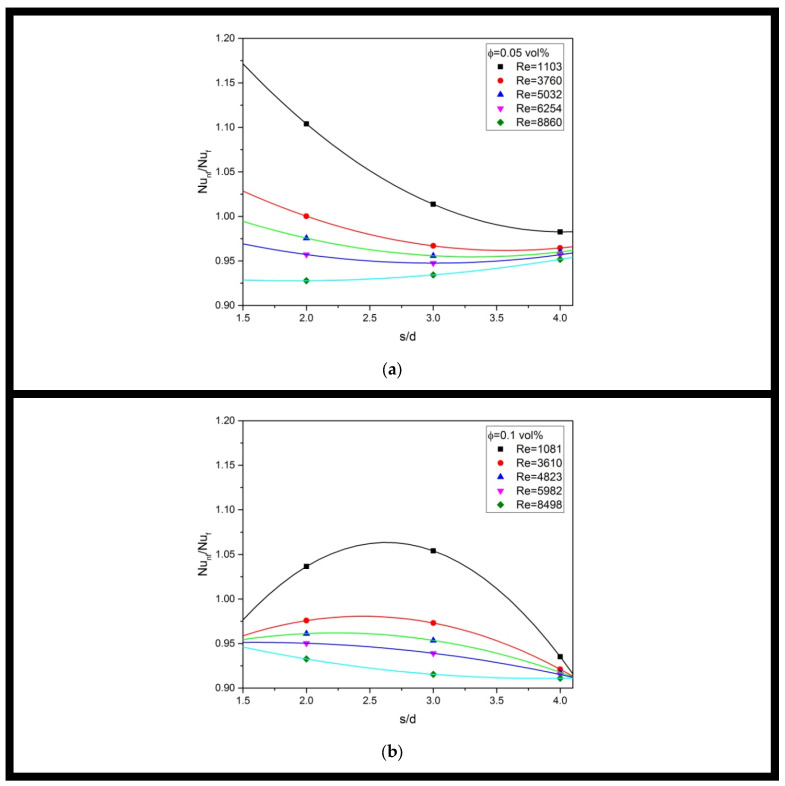
Nusselt Number Enhancement Ratio of MgO-Water Nanofluid versus Jet-to-Jet Spacing with Different Re at (**a**) ϕ = 0.05 vol%; (**b**) ϕ = 0.1 vol%; (**c**) ϕ = 0.15 vol%.

**Table 1 materials-16-03942-t001:** Important parameter ranges of experimental studies of water-based nanofluids jet impingement.

Author	NP	Concentration	Jet Number	Jet’s Height	Jets’ Spacing	Reynold Number	Enhancement Ratio ^a^
(A) Submerged jet impingement							
[4]	Al_2_O_3_	ϕ = 5	N_jet_ = 1	2 ≤ H ≤ 10	-	1700 ≤ Red ≤ 20,000	h_nf_/hf = 1.72 ^b^ ($\dot{m}$)
[5]	Al_2_O_3_	ϕ = 2.8 and 6	N_jet_ = 1	2 ≤ H ≤ 10	-	3800 ≤ Red ≤ 88,000	0.53 ≤ h_nf_/hf ≤ 1.35 (Re_d_)
[6]	Al_2_O_3_	2 ≤ ϕ ≤ 6	N_jet_ = 1	H = 2 and 3	-	500 ≤ ReH ≤ 901	1.06 ≤ Nu_nf_/Nuf ≤ 2.4 (Re_H_)
[7]	Cu	1.5 ≤ ϕ ≤ 3	N_jet_ = 1	2 ≤ H ≤ 6	-	2000 ≤ Red ≤ 16,000	1 ≤ Nu_nf_/Nuf ≤ 1.25 (Re_d_)
[8]	Cu	1.5 ≤ ϕ ≤ 3	N_jet_ = 1	1 ≤ H/d ≤ 3	-	2000 ≤ Red ≤ 16,000	1.13 ≤ h_nf_/hf ≤ 1.54 (Re_d_)
[9]	Ag	0.02 ≤ φ ≤ 0.13	N_jet_ = 7 ^d3^	H = 3	s = 7	200 ≤ Red ≤ 800	1.08 ≤ Nu_nf_/Nuf ≤ 1.81 (Re_d_)
(B) Free surface jet impingement							
[10]	Al_2_O_3_	φ = 6.6 and 10	N_jet_ = 1	H = 50	-	2500 ≤ Re_d_i ≤ 24,000	0.82 ≤ Nu_nf_/Nuf ≤ 2.21 (Re_d_)
[11]	Al_2_O_3_	φ = 6.6 and 10	N_jet_ = 1	H = 50	-	2500 ≤ Re_d_i ≤ 24,000	0.96 ≤ Nu_nf_/Nuf ≤ 2.06 (Re_d_)
[12]	Al_2_O_3_	0.0198 ≤ φ ≤ 0.08	N_jet_ = 1	H = 48	-	4200 ≤ Red ≤ 8200	1.04 ≤ h_nf_/hf ≤ 1.48 (Re_d_)
[13]	Al_2_O_3_	1 × 10^−4^ ≤ ϕ ≤ 0.01	N_jet_ = 1	H = 10	-	-	0.8 ≤ h_nf_/hf ≤ 1.27 ($q^{\prime \prime }$)
[14]	Al_2_O_3_	2 ≤ ϕ ≤ 10	N_jet_ = 1	H/d = 3	-	3000 ≤ Red ≤ 32,000	1.02 ≤ Nu_nf_/Nuf ≤ 1.67 (Re_d_)
[15]	SiO_2_	1 ≤ ϕ ≤ 3	N_jet_ = 1	2 ≤ H/d ≤ 5	-	8000 ≤ Red ≤ 13,000	1.12 ≤ h_nf_/hf ≤ 1.76 (Re_d_)
[16]	Al_2_O_3_	0.5 ≤ ϕ ≤ 2	N_jet_ = 1	2 ≤ H/d ≤ 5	-	8000 ≤ Red ≤ 13,000	1.10 ≤ h_nf_/hf ≤ 1.75 (Re_d_)
[17]	ZnO	0.02 ≤ ϕ ≤ 0.1	N_jet_ = 1	2 ≤ H/d ≤ 7.5	-	2192 ≤ Red ≤ 9241	1.17 ≤ h_nf_/hf ≤ 1.58 (Re_d_)
[18]	CuO	0.1 ≤ ϕ ≤ 0.3	N_jet_ = 1	40 ≤ H ≤ 70	-	1000 ≤ Red ≤ 8000	0.65 ≤ Nu_nf_/Nuf ≤ 2.24 (Re_d_)
[19]	SiO_2_	0.5 ≤ ϕ ≤ 8.5	N_jet_ = 1	0.5 ≤ H/d ≤ 8	-	2148 ≤ Red ≤ 40,004	0.96 ≤ Nu_nf_/Nuf ≤ 2.51 ^c^ (Re_d_)
[20]	Cu_2_O	0.03 ≤ ϕ ≤ 0.07	N_jet_ = 1	-	-	7330 ≤ Red ≤ 11,082	1.2 ≤ h_nf_/hf ≤ 1.31 (Re_d_)
[21]	ZnO	0.1 ≤ ϕ ≤ 0.5	N_jet_ = 1	2 ≤ H/dh ≤ 8	-	5000 ≤ Re_d_h ≤ 17,500	1.3 ≤ Nu_nf_/Nuf ≤ 2.27 (Re_dh_)
[22]	Al_2_O_3_	2 ≤ ϕ ≤ 6	13 < N_jet_ < 81 ^d1^	2 ≤ H/d ≤ 17	3 ≤ s/d ≤ 7	2000 ≤ Red ≤ 10,000	2.13 ≤ Nu_nf_/Nuf ≤ 5.98 (Re_d_)
[23]	Cu	0.17 ≤ ϕ ≤ 0.68	N_jet_ = 5 ^d4^	H = 15	s = 4	4000 ≤ Red ≤ 10,000	0.93 ≤ h_nf_/hf ≤ 1.15 (Q_v_)
[24]	Al_2_O_3_	0.5 ≤ ϕ ≤ 4.9	9 < N_jet_ < 17 ^d2^	H/d = 12.33	3 ≤ s/d ≤ 7	1280 ≤ Red ≤ 6500	0.74 ≤ Nu_nf_/Nuf ≤ 1.04 (Re_d_)
[25]	Al_2_O_3_	0.05 ≤ ψ ≤ 0.25	N_jet_ = 2	4 ≤ H/d ≤ 7	-	400 ≤ Red ≤ 2000	1.21 ≤ Nu_nf_/Nuf ≤ 3.14 (Re_d_)
[26]	Al_2_O_3_	5 ≤ ϕ ≤ 10	N_jet_ = 9 ^d5^ and 10 ^d6^	10 ≤ H/d ≤ 40	s/d = 7	5600 ≤ Red ≤ 33,610	1.38 ≤ Nu_nf_/Nuf ≤ 3.3 (Re_d_)

Remarks: ^a^ The enhancement ratio ranges are estimated based on reported data points from the respective papers. The comparison basis is shown in the bracket beside the range. ^b^ Only one enhancement ratio is provided by the author, so no enhancement ratio range can be reported. ^c^ This enhancement ratio range is estimated based on the author’s proposed correlation equation. ^d1^, ^d2^, ^d3^, ^d4^, ^d5^, and ^d6^ The nozzle arrangement can be referred to as Figure 1a, Figure 1b, Figure 1c, Figure 1d, Figure 1e, and Figure 1f, respectively.

**Table 2 materials-16-03942-t002:** Thermophysical property of MgO nanopowder [46].

Properties	Unit	Value
ρ	kg/m^3^	3585
Cp	J/(kg K)	903
k	W/(mK)	48.4

**Table 3 materials-16-03942-t003:** Thermal-physical properties of MgO-water nanofluids at 24 °C.

Volume Fraction (%)	k (W m^−1^ k^−1^)	μ (Pa s)	ρ (kg/m^3^)	Cp (J kg^−1^ k^−1^)
0	0.609	0.000912	997.32	4180.14
0.05	0.620	0.000950	998.61	4174.26
0.1	0.623	0.000992	999.91	4168.39
0.15	0.629	0.00102	1001.20	4162.54

**Table 4 materials-16-03942-t004:** Uncertainty of the Measured Quantity.

*w_i_*	Instrument	Smallest Reading	U*_w_*_i_
d	Mitutoyo Digital Caliper	1.5 mm	±1.6%
T	PicoLog TC-08	24 °C	±0.2%
ΔT	ΔT = T_s_ − T_jet_	1.2 °C	±3%
dT/dz	-	148 °C/m	±0.8%
Q_v_	FS200A flow sensor	1.6 LPM	±2.15%
m_f_	WLC mass balance	200 g	±0.05%
m_p_	MS104TS mass balance	0.36 g	±0.03%
k	KD2 PRO analyzer	0.6 W/(m k)	±5%
μ	SV-10 viscometer	0.912 mPa s	±1%

**Table 5 materials-16-03942-t005:** Grid independence and grid convergence index (GCI) results for all models under u_jet_ = 5.31 m/s using water as coolant.

Models	Grid Level	Number of Elements	Nu_D_	Deviation between Grid Levels	GCI
s/d = 4	Coarse	157,114	1664.297	-	-
	Fine	385,543	1670.69	0.39%	0.24%
	Very fine	782,680	1712.388	2.50%	3.3%
s/d = 3	Coarse	156,410	1496.977	-	-
	Fine	388,250	1516.461	1.30%	1.2%
	Very fine	771,654	1558.061	2.74%	4.6%
s/d = 2	Coarse	159,450	1328.039	-	-
	Fine	398,042	1294.999	−2.50%	0.14%
	Very fine	777,894	1290.181	−0.37%	0.6%

## Data Availability

Data are available upon request from the corresponding author. These data are not commercially available due to privacy issues.

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
