# Peer review of "Experimental and Numerical Investigation of Flow Structure and Heat Transfer Behavior of Multiple Jet Impingement Using MgO-Water Nanofluids"

_materials, 2023, doi:10.3390/ma16113942_

Round 1

Reviewer 1 Report

In this paper, the authors carried out an experimental and numerical investigation to study the flow structure and heat transfer of multiple jet impingement using MgO-water nanofluids. A 3-D numerical analysis using ANSYS Fluent 16 with SST k-ω turbulent model was presented. The single phase model is adopted to predict the thermo-physical nanofluid properties. By the experimental results the authors shown that a nanofluid can provide a heat transfer enhancement at a small jet-to-jet spacing using a high particle volume fraction under a low Reynolds number; otherwise, an adverse effect on heat transfer may occur. By the numerical results, they have shown that the single phase model can predict the heat transfer trend of multiple jet impingement using nanofluids correctly, but with significant deviation from experimental results because it cannot capture the effect of nanoparticles.

The stated problem is interesting. Results in this paper are useful to the researchers in the field of heat transfer enhancement.

Observations:

1). In the Abstract and in whole text, instead of “Reynold” must be Reynolds (after the name of scientist Osborne Reynolds 1842-1912).

2). Line 167. The authors wrote “Uy, is calculated by 167 Equation 8”. But, the equation (8) contains the parameter U_f. Check and correct.

3). Nomenclature. Verify the dimension unit of mass “m”. It is “g” or “Kg” ? The dimension of some parameters is not written (for example the velocity “u”).

Author Response

Dear Reviewer,

Please find the attached response.

Best regards

Reviewer 2 Report

Paper: materials-2316579

Title: Experimental and Numerical Investigation of Flow Structure and Heat Transfer Behaviour of Multiple Jet Impingement Using MgO-Water Nanofluids

The present manuscript presents both experimental and numerical studies related with flow structure and heat transfer behaviour of multiple jet impingement using MgO-water nanofluids with a 3x3 inline jet array at a nozzle-to-plate distance of 3 mm. The numerical approach includes 3-D effects by using ANSYS Fluent with SST k-ω turbulent model. The experimental results have pointed that nanofluids can provide a heat transfer enhancement at a small jet-to-jet spacing using a high particle volume fraction under a low Reynolds number. On the other hand, numerical results have showed that single phase model is able to predict the heat transfer trend of multiple jet impingement using nanofluids correctly, however, there is significant deviation from experimental results because it cannot capture the effect of nanoparticles. The research topic is relevant to be published as material´s paper. However, I have some comments/recommendations addressed to the authors, before the manuscript to be properly accepted.

(i) Moderate English changes required.

(ii) In “Abstract”, the main contribution of the present work must be clarified and the present state of the art concerning the investigated topic must also be introduced and contextualized. What does justify the intended publication in materials?

(iii) On page 2, line 71, the authors have commented the following: “The research on nanofluids' multiple jet impingement cooling is still new and not well assessed.” However, the “Introduction” of the present manuscript must help the reader understand why a research is important and what it is contributing to the research field. Put in other words, the context of the present “Introduction” (a) must state the motivation for doing a research and what it will contribute to the research field; (b) must present a brief overview of the current state of research; (c) must present some more detailed information on the specific topic of the research; and (d) must present a description of the exact question or hypothesis that the paper will address. Thus, it is necessary that authors present more technical discussions concerning these key points including examples of past models through a critical review. As consequence, the contributions of the manuscript can better explained and contextualized in this crucial part of the manuscript. The adopted methodologies in the present research need to be introduced and justified too. So, I would like to recommend a new wording including a critical literature review more comprehensive that justify the intended publication in materials.

(iv) Before starting the section 2.1“Nanofluids Synthesis and Thermo-physical Properties”, the authors should present specific comments concerning the assumed hypothesis, governing equations (which are they?) and boundary conditions for the chosen problem. A figure could also be properly linked with the idea of the general formulation of the problem. In fact, the manuscript is poor concerning Mathematical Approach of the physical problem.

(v) How is computed the local energy dissipation? Please, introduce the concept of turbulent viscosity and contextualize it into the adopted methodology.

(vi) Section 8 is very extensive and confuse; there are lots of information. The section 2.8 “Computational Details” should be Section 3, for example. So, it is recommend separating sections Mathematical Formulation, Experimental Method and Numerical Method.

(vii) If the authors also utilize a numerical method, it is important to include some explanation about implementation of the present method algorithm.

(viii) The manuscript does not make clear the buoyancy effects influence. Are they important?

(ix) Equation (6) needs to be better introduced and explained.

(x) The Prandtl number effect was not explained. For instance, what is the number Prandtl influence on results of Figure 6.  A discussion about Richardson number effect is welcome too.

(xi) In “Conclusions”, it is necessary to include more comments with respect the numerical results behaviour as compared as previous works (specially, numerical data). The present study has showed that numerical approach is not able to capture the effect of nanoparticles. In closing, it is important to complete the manuscript with perspectives for a future research. Finally, the main contribution of the present manuscript should be clarified aiming to justify its publication in materials.

III - Recommendation for the materials´ editor

In my opinion, the present manuscript needs attend all topics above presented. Upon consideration of all points above, I think the paper could be considered for publication in materials.

Author Response

(The authors gave the same response as above.)

Reviewer 3 Report

In this study, authors performed both numerical and experimental work for flow and heat transfer of multiple jet impingement. They used MgO-water nanofluid with loading amount up to 0.15% while Re value was considered between 1000 and 10000. They used commercial CFD code based on finite volume method and SST k-ω turbulent model was adopted.

This paper is interesting and important outcomes were achieved. However, following major comments should be considered:

1) The solution methodology is weak; it should be improved.

2) All solver setting should be given.

3) Resolution of figures 1,2,3, 4 may be increased.

4) Please remove the bold boxes in Figures 3, 4

5) Grid independence results should be given in Tabular or graph

6)Following papers are relevant

International Journal of Heat and Mass Transfer 203, 123764, 2023

Nanomaterials 13 (3), 500, 2023

7) Some abbreviations are missing in the Nomenclature.

8) The validation results may be given.

9) Conclusion can be expressed in the form of bullet points.

10) Some spelling errors should be avoided.

Author Response

(The authors gave the same response as above.)

Round 2

Reviewer 2 Report

Paper: materials-2316579-v2

Title: Experimental and Numerical Investigation of Flow Structure and Heat Transfer Behaviour of Multiple Jet Impingement Using MgO-Water Nanofluids

The original text of the manuscript has been satisfactorily revised by authors. In my opinion, the manuscript can be published as materials’ paper.

Reviewer 3 Report

It may be accepted in this format.